

# Managing slow-moving item: a zero-inflated truncated normal approach for modeling demand

Fernando Rojas[1,2], Peter Wanke[3], Giuliani Coluccio[4], Juan Vega-Vargas[4] and Gonzalo F. Huerta-Canepa[5]

[1] Micro-BioInnovation Center, Universidad de Valparaíso, Valparaíso, Chile
[2] School of Nutrition and Dietetics, Universidad de Valparaiso, Chile, Valparaíso, Chile
[3] COPPEAD, Universidade Federal do Rio de Janeiro, Rio de Janeiro, Brazil
[4] Department of Industrial and Systems Engineering, Faculty of Engineering, Universidad de Tarapacá, Arica, Chile
[5] Faculty of Engineering and Sciences, Universidad Adolfo Ibáñez, Viña del Mar, Chile

Corresponding author
Fernando Rojas, fernando.rojas@uv.cl

## ABSTRACT

This paper proposes a slow-moving management method for a system using of intermittent demand per unit time and lead time demand of items in service enterprise inventory models. Our method uses zero-inflated truncated normal statistical distribution, which makes it possible to model intermittent demand per unit time using mixed statistical distribution. We conducted numerical experiments based on an algorithm used to forecast intermittent demand over fixed lead time to show that our proposed distributions improved the performance of the continuous review inventory model with shortages. We evaluated multi-criteria elements (total cost, fill-rate, shortage of quantity per cycle, and the adequacy of the statistical distribution of the lead time demand) for decision analysis using the Technique for Order of Preference by Similarity to Ideal Solution (TOPSIS). We confirmed that our method improved the performance of the inventory model in comparison to other commonly used approaches such as simple exponential smoothing and Croston's method. We found an interesting association between the intermittency of demand per unit of time, the square root of this same parameter and reorder point decisions, that could be explained using classical multiple linear regression model. We confirmed that the parameter of variability of the zero-inflated truncated normal statistical distribution used to model intermittent demand was positively related to the decision of reorder points. Our study examined a decision analysis using illustrative example. Our suggested approach is original, valuable, and, in the case of slow-moving item management for service companies, allows for the verification of decision-making using multiple criteria.

## INTRODUCTION AND LITERATURE REVIEW

Intermittent demand occurs when the demand per unit of time (DPUT) for products, parts, or pieces in some periods is zero (*Syntetos et al., 2016*). This type of DPUT is

often considered as a random variable due to its stochastic nature (*Cattani, Jacobs & Schoenfelder, 2011*). Intermittent DPUT a common occurrence for service and commercial companies that supply parts to industry sectors such as aerospace (*Ayu Nariswari, Bamford & Dehe, 2019*), automotive (*Zhang & Xiaofeng, 2017*), information technology (*Antosz & Ratnayake, 2019*), and military (*Babai, Syntetos & Teunter, 2014*). This particular frequency in demand could impact different company strategies (*Pavlas et al., 2017*; *Devika et al., 2016*), such as the use of inventory models (*Jonkman, Barbosa-Póvoa & Bloemhof, 2019*). Inventory models reduce costs and establish optimal stock levels in order to meet the demand for components and final products for customers (*Rojas et al., 2019*). However, in models predicting zero demand, its parameters are not calculated the same because demand is characterized by intermittency, and the particularities of the intermittent demand need to be considered to develop more accurate inventory models (*Gregersen & Hansen, 2018*). Inventory models that consider this type of demand are called slow-moving items (*Hahn & Leucht, 2015*). Intermittent DPUT is also characterized by high variability across the non-zero values that compose the demand, requiring precise forecast models to be used in inventory models (*Kim & Kim, 2016*).

Traditional inventory models are based on a fixed demand (*Teixeira, Lopes & Figueiredo, 2018*). However, updated models should factor in the uncertainty of demand (*Aloulou, Dolgui & Kovalyov, 2014*). The uncertainty of DPUT and lead time demand (LTD) is a crucial aspect in supply chain management (*Khosravi et al., 2018*). One of the most commonly used models in service company supply is the $Q, r$ model, which is a continuous review inventory model system (*Ponte et al., 2018*; *Wen et al., 2020*). This model reorders a fixed quantity ($Q$) in a single period and makes a purchase or production order under an on-hand inventory level named the reorder point ($r$). We only need the average DPUT forecast to determine Q, but knowing the complete LTD distribution is required to determine $r$. In order to facilitate calculations, a normal distribution for both DPUT and LTD is usually assumed (*Johnson, Kotz & Balakrishnan, 1994*). Nevertheless, other distributions used to model DPUT, lead time (LT), and LTD can provide more accurate inventory modeling. Table 1 in *Cobb, Rumí & Salmerón (2013)* provides a summary of the distributions used in stochastic inventory models.

Although the normal distributed $Q, r$ model is competent at inventory management, applying it in models of intermittent demand could produce biased results. Normal distribution does not work in intermittent demand modeling because it is difficult to predict empirical variation produced between non-zero and zero DPUTs. To address this problem and to treat this kind of data, several forecasting methods have been developed such as the: simple statistical smoothing methods, Croston's variant of exponential smoothing (*Croston, 1972*; *Kourentzes, 2014*), and bootstrap methods (*Willemain, Smart & Schwarz, 2004*; *Ewbank et al., 2020*). Among these, the bootstrap methods have shown the best precision in updating the LTD and distributing the underlying probability of non-zero values. However, all of these methods have problems in providing precise estimates, given that the overdispersion of data coming from an intermittent DPUT negatively affects parameters estimation and truncated probability distributions. (*Zeileis, Kleiber & Jackman, 2008*) and (*Yang, 2012*) improved parameter estimation using the maximum likelihood

method that combines variable counts, zero mass points called hurdle models, truncated counts on the left, and censored counts on the right called zero-adjusted models. These distributions may be especially suitable for intermittent demand because of their ability to explicitly model non-zero and zero demand cases. Statistical mixture distribution models can describe, estimate and simulate these types of data (*Bethea, 2018*). Here we highlight the $Q, r$ model and its use in slow-moving items. Using zero-inflated statistical distribution modeling intermittent demand, appears promising, but its inventory modeling performance has not been adequately evaluated or compared to other methods (*Ünlü, 2011*).

Our objective was to propose slow-moving item management model that uses the statistical distribution of mixture, zero-inflated truncated normal (ZITNO), where the Normal distribution component's domain is defined only in Real positive, to model intermittent DPUT forecasting with non-zero values and the LTD by means of a zero-inflated truncated normal sum (ZITNOsum). We examined their performance using a continuous review with a shortage inventory model that included total costs of inventory, fill-rate, the quantity of inventory shortage per cycle, and the statistical distribution of LTD. In 'Background', we describe the background of our proposal. We explain how to generate and implement several intermittent DPUT forecast models, including one predicting LTD when LT is constant. We use the following statistical distributions: ZITNO / ZITNO sum, Simple exponential smoothing / Simple exponential smoothing sum, and Croston's / Croston's sum, a Simple exponential smoothing method variant. 'Numerical Experiments and Illustrative Example' is divided into three parts. 'Evaluating inventory model performance measures using TOPSIS' shows the numerical experiments we conducted to show the benefits of our proposed model. We compared different inventory models with multi-criteria decisions using the Technique for Order of Preference by Similarity to Ideal Solution (TOPSIS) when the statistical distributions modeled the DPUT and LTD. In 'Effect of variability and intermittent DPUT with ZITNO statistical distribution on total costs and $Q$ and $r$ decisions', we determine the parameter of variability and how the proportion of zeros contained in the ZITNO statistical distribution affected total costs, $Q$ and $r$. In 'Analyzing real data using an illustrative example', we show a decision analysis using an illustrative example. Finally, in 'Discussion', we discuss our findings, the limitations of the study, and our conclusions.

## BACKGROUND

### Forecasting intermittent DPUT and LTD

In this section, we show how to forecast an item's complete LTD distribution. Let $Y$ be an random variable of the DPUT. This DPUT is intermittent, meaning that , sometimes it is zero and sometimes it is not. This creates great variability in the data. Let $S$ be the LTD, which corresponds to a random sum that is expressed as:

$$S = \sum_{t=T+1}^{T+K} Y_t, \tag{1}$$

where $K$ is the fixed LT used for forecasting LTD, with mean $E(K) = \mu_K$ and variance $Var(K) = \sigma_K^2 = 0$. We consider $K$ fixed when calculating LTD random variables using the formula $\{Y_{T+1} + \cdots + Y_{T+K}, k \in \mathbb{N}\}$. We calculate the LTD mean using the formulas $E(\{Y_{T+1} + \cdots + Y_{T+K}\}) = \mu_S$ and variance $Var(\{Y_{T+1} + \cdots + Y_{T+K}\}) = \sigma_S^2$.

(*Willemain, Smart & Schwarz, 2004*) found that intermittent DPUT is often executed in strokes with longer sequences of zeros and other non-zero values. For this reason, it is possible to use a pattern of autocorrelation and a Markov process of two first-order states can be used to forecast this random variable with temporal sequence. Starting with a prediction of the sequence of zero and non-zero values during the $K$ LT periods, these forecasts are conditioned to determine whether the last demand, $Y_T$, is zero or non-zero. Using the counts of a historical or simulated demand time series, it possible to estimate the probabilities of state transitions (*Mosteller & Tukey, 1977*). Scientific computing and simulation overlay play fundamental roles in generating knowledge and studying decision-making (*Salvatier, Wiecki & Fonnesbeck, 2016*). It is therefore necessary to assign numeric values to non-zero predictions that cannot be based on unrealistic bootstraps, particularly those with poorly estimated LTD distribution tails made with values from the same historical data set. This problem is solved with jittering, defined as adding some random variations assumed with normal distribution in order to allow the use of values closer to the historical data. We adapted this method to generate a LTD jitter that is able to occupy an intermittent DPUT in a simulation approach for slow-moving items. A summary of this approach can be found in Algorithm 1. The execution of Algorithm 1 requires R software, a free software for statistics and graphs that is used across the international scientific community, and can be consulted in the codes attached to this work with the name of "Jitter.R". (*Rojas, 2016*) used this software in supply models and programmed an R code in a generalized linear model (GLM) environment. This allowed them to generate a sequence of random values following the statistical distribution ZITNO, as well as to estimate the parameters of this statistical distribution, among other functionalities. The `rmarkovchain` command of R package `markovchain` generates a random sequence of zero/non-zero markers of a known length for an random variable using an estimated transition probability matrix (*Spedicato, 2015*).

## Modeling DPUT and LTD with a constant LT

In this subsection we show three statistical distributions that can be occupied when modeling of random variable DPUT, and three modeling distributions of the LTD, which is the sum of this random variable in a constant LT. These three pairs of statistical distributions are: Simple exponential smoothing /Simple exponential smoothing sum, Croston's / Croston's sum and ZITNO / ZITNOsum. For all cases, the following models assume that DPUT forecast are generated from a time $T + 1, \ldots, T + m$. For the LTD, it is assumed that LT is constant ($\mu_K$). DPUT and LTD forecasts are generated from Algorithm 1.

### *Simple exponential smoothing / Simple exponential smoothing sum*

This approach assume that LTD follows a normal statistical distribution and a fixed LT. The LTD mean and variance are calculated as follows:

---

**Algorithm 1** Generating intermittent DPUT and LTD forecasts for use in a slow-moving item inventory management model

---

1:  Generate a random sample of intermittent DPUTs of ZITNO statistical distribution of length $n$ and fixed and known parameters.

2:  Estimate the transition probability matrix of the zero and non-zero DPUTs of $n$ generated in Step 1.

3:  Conditional on the last observed demand, generate a random sequence of zero/non-zero DPUT markers of known length using the transition probability matrix estimated in Step 2.

4:  Replace every non-zero state marker with a numerical value sampled at random, with replacement, from the original set of observed non-zero DPUTs generated in Step 1.

5:  Estimate the parameters of normal distribution adjusted to the non-zero values of the random sample with replacement achieved in Step 4.

6:  Generate a "jitter" of the non-zero DPUT values, replacing the non-zero markers generated in Step 3 with random numbers generated from the normal statistical distribution with estimated parameters in Step 5.

7:  From the sample "jitter" obtained in Step 6, sum the values over the horizon of a constant LT to get LTD forecast values.

---

(i) Considering the mean level of DPUT as $\mu_{SES}$, and estimate using

$$\mu_{SES} = \frac{1}{T+m} \sum_{t=T+1}^{T+m} (\gamma y_t + (1-\gamma)\mu_{t-1_{SES}}),$$

where $\gamma$ is a smoothing constant between 0 and 1, selected to minimize $\sum_{t=T+1}^{T+m}(y_t - \mu_{SES})^2, t = T+1,\ldots,T+m$. To initialize the smoothing, we can use the average of the first two demands $\mu_o = \frac{y_1+y_2}{2}$.

(ii) The DPUT variance with this approach can be calculated from:

$$\mathrm{Var}(Y) = \sigma_{SES}^2 = \frac{1}{T+m} \sum_{t=T+1}^{T+m} (y_t - \mu_{SES})^2, \forall \gamma.$$

The mean of $K$ demands over the LT ($\mu_{S_{SES}}$) is given by

$$\mu_{S_{SES}} = \mu_K \mu_{SES},$$

and the variance of $K$ demands over the lead time ($\sigma_{SES}^2$) is calculated using one-step ahead forecast difference between the actual DPUT data and the mean lag, using the expression $\sigma_{SES}^2 = \frac{1}{m}\sum_{t=T+1}^{T+m}(y_t - \mu_{t-1_{SES}})^2$. The variance of the LTD distribution ($\sigma_{S_{SES}}^2$) can be estimated as:

$$\sigma_{S_{SES}}^2 = \sigma_{SES}^2 \mu_K.$$

### Croston's / Croston's sum variant of Simple exponential smoothing method

Croston's approach considers the DPUT mean using exponential smoothing that is separate from:

(i) the mean intervals of data conformed between non-null demands (here, the smoothed estimate is denoted by $I_t$) and (ii) the mean sizes of these intervals (here, the smoothed estimate is denoted by $S_t$).

In addition, $q$ is the time interval since the last non-zero demand. Croston's approach can be described as follows:

if $Y = 0$, then

$$S_t = S_{t-1}$$
$$I_t = I_{t-1}$$
$$q = q+1, \tag{2}$$

else

$$S_t = \gamma Y_t + (1-\gamma)S_{t-1}$$
$$I_t = \gamma q + (1-\gamma)I_{t-1}$$
$$q = 1. \tag{3}$$

The combination of the size and interval estimates from Eqs. (2) and (3), the DPUT mean can be expressed as:

$$\mu_{CROST} = \frac{1}{T+m} \sum_{t=T+1}^{T+m} \left(\frac{S_t}{I_t}\right)$$

These estimates update whenever a demand non null realization occurs. When a demand occurs during the same review interval, Croston's approach is identical to conventional exponential smoothing, where $S_t = \mu_{t_{CROST}}$.

To initialize Croston's approach, we use the time until the first event and the size of the first event.

The DPUT variance when using this method can be expressed as:

$$\mathrm{Var}(Y) = \sigma^2_{CROST} = \frac{1}{T+m} \sum_{t=T+1}^{T+m} (y_t - \mu_{CROST})^2.$$

Croston's method also considers LTD with a constant LT and normal statistical distribution. The mean is expressed as:

$$\mu_{S_{CROST}} = \mu_K \mu_{CROST},$$

and the variance is calculated as:

$$\sigma^2_{S_{CROST}} = \sigma^2_{CROST} \mu_K.$$

### Zero inflated truncated normal / zero inflated truncated normal sum

In this model, we assumed that DPUT has a ZITNO distribution and LTD has a ZITNOsum distribution with a constant LT. We estimated the mean level of DPUT ($\mu_{ZITNO}$) and its variance ($\sigma^2_{ZITNO}$) using:

$$E(Y) = \mu_{ZITNO} = (1-v) \int_{R_{y>0}} y\phi\left(\frac{y - \mu_{NO}}{\sigma_{NO}}\right) \frac{1}{\sigma_{NO}} dy; \text{ with } y > 0,$$

and

$$\text{Var}(Y) = \sigma^2_{ZITNO} = \frac{1}{T+m} \sum_{t=T+1}^{T+m} (y_t - \mu_{ZITNO})^2,$$

respectively, where $\mu_{NO}$ and $\sigma_{NO}$ are mean parameters and the standard deviation (SD) of a normal distribution of subset $y > 0$. Note that $Y$ forecasting length measures from $T+1$ to $T+m$. On the other hand, the expected LTD value ($\mu_{S_{ZITNOsum}}$) and its variance ($\sigma^2_{S_{ZITNO}}$) under ZITNOsum distribution is calculated by:

$$\mu_{S_{ZITNOsum}} = \mu_K \mu_{ZITNO},$$

and

$$\sigma^2_{S_{ZITNOsum}} = \sigma^2_{ZITNO} \mu_K,$$

respectively.

## Intermittent DPUT and LTD in the $Q, r$ model with shortage

In $Q, r$ model with shortage, the expected annual total cost is the a sum of:

(i) the product of the expected product stock quantity (in units) and holding cost per product unit per year (HC);

(ii) the product of the expected number of orders per year and the ordering cost(OC), and finally

(iii) the product of the unit punishment cost (SC) per units of the item in short supply, the expected number of orders per year, and the expected number of units of shortage product per year, which is a function of the reorder point ($S(r)$).

We assumed that the organization maintains intermittent demand every day of the year (365).

The expected total cost per year (TC) in the $(Q, r)$ model can be expressed as:

$$\text{TC} = G(Q, r) = \left(\frac{Q}{2} + r - \mu_S\right)\text{HC} + \frac{365\mu}{Q}\text{OC} + S(r)\frac{365\mu}{Q}\text{SC}, \qquad (4)$$

where $\mu$ and $\mu_S$ values (note that the sequence of values of $Y$ forecast values from $T+1$ to $T+m$, and that the DPUT sum needed to forecast the LTD is calculated using LT $= K$), can be calculated according to the probabilistic modeling showed in Tables 1 and 2. For diverse DPUT and LTD statistical distributions, see the probability density functions (PDFs), cumulative distribution functions (CDFs) and parameters in Tables 1 and 2(*Hadley & Whitin, 1963*; *Johnson & Montgomery, 1974*; *Silver, Pyke & Peterson, 1998*).

**Table 1 Modeling DPUT.**

| Distribution | PDF | CDF | Parameters |
|---|---|---|---|
| ZITNO | $(1-\nu)\phi(\frac{y-\mu_{NO}}{\sigma_{NO}})\frac{1}{\sigma_{NO}}1_{(y>0)}$ $+\nu 1_{(y=0)}$ | $\nu+(1-\nu)\Phi(\frac{y-\mu_{NO}}{\sigma_{NO}})$ | $\mu_{NO}\in\mathbb{R},\sigma_{NO}>0$ $,0<\nu<1$ |
| Simple exponential smoothing | $\phi(\frac{y-\mu_{SES}}{\sigma_{SES}})\frac{1}{\sigma_{SES}}$ | $\Phi(\frac{y-\mu_{SES}}{\sigma_{SES}})$ | $\mu_{SES}\in\mathbb{R},\sigma_{SES}>0$ |
| Croston's | $\phi(\frac{y-\mu_{CROST}}{\sigma_{CROST}})\frac{1}{\sigma_{CROST}}$ | $\Phi(\frac{y-\mu_{CROST}}{\sigma_{CROST}})$ | $\mu_{CROST}\in\mathbb{R},\sigma_{CROST}>0$ |

**Table 2 Modeling LTD.**

| Distribution | PDF | CDF | Parameters |
|---|---|---|---|
| ZITNOsum | $(1-\nu_s)\phi(\frac{s-\mu_{NO}\mu_k}{\sigma_{NO}(\mu_k)})\frac{1}{\sigma_{NO}(\mu_k)}1_{(s>0)}$ $+\nu_s 1_{(s=0)}$ | $\nu_s+(1-\nu_s)\Phi(\frac{s-\mu_{NO}\mu_k}{\sigma_{NO}(\mu_k)})$ | $\mu_{NO}\mu_k\in\mathbb{R},\sigma_{NO}(\mu_k)>0$ $,\mu_k\geqq 2,0<\nu_s<1$ |
| Simple exponential smoothing sum | $\phi(\frac{s-\mu_{SES}\mu_k}{\sigma_{SES}\mu_k})\frac{1}{\sigma_{SES}\mu_k}$ | $\Phi(\frac{s-\mu_{SES}\mu_k}{\sigma_{SES}\mu_k})$ | $\mu_{SES}\mu_k\in\mathbb{R},\sigma_{SES}\mu_k>0$ |
| Croston's sum | $\phi(\frac{s-\mu_{CROST}\mu_k}{\sigma_{CROST}\mu_k})\frac{1}{\sigma_{CROST}\mu_k}$ | $\Phi(\frac{s-\mu_{CROST}\mu_k}{\sigma_{CROST}\mu_k})$ | $\mu_{CROST}\mu_k\in\mathbb{R},\sigma_{CROST}\mu_k>0$ |

In this formula, $365\mu$ corresponds to the expected annual demand, to express the annual costs referred to in Eq. (4). However, $\mu_S$ does not require this transformation. $Q$ and $r$ correspond to lot size decision variable to order and reorder points, respectively. $S(r)$ is the expected shortage per cycle calculated as:

$$S(r)=\int_r^{s_{max}}(s-r)f_S(s)\,ds, \tag{5}$$

where $s_{max}$ is the maximum LTD value and the LTD PDF is denoted by $f_S(\cdot)$. This expression can also be calculated using different assumptions LTD PDF assumptions shown in Table 2. For any statistical distribution of DPUT and LTD, we can solver Eq. (4) using an iterative method, considering an initial solution of $Q=\sqrt{\frac{2C_o\mu}{C_h}}$ (*Nahmias, 2001*). Here, the probability of obtaining a stockout when given the complement of the CDF ($F_s(r)$) can be expressed as:

$$1-F_s(r)=\frac{QHC}{\mu SC}.$$

To calculate the argument of this function ($r$), we applied this inverse function:

$$r=F_s^{-1}(1-\frac{QHC}{\mu SC}). \tag{6}$$

To estimate the expected vale of $S(r)$ function in Eq. (5) and to find the optimum lot size, we used:

$$Q=\sqrt{\frac{2\mu(OC+S(r))}{HC}}. \tag{7}$$

We repeated Eqs. (6) and (7) until we reached a value of variation smaller than a previously established minimum threshold. We were then able to calculate lower $Q*,r*$ values than Eq. (4).

## Measuring inventory model performance

In this Section, we will define some previously proposed general performance measures to evaluate a continuous review inventory model. These measures are applicable for the DPUT and LTD modeling shown in Tables 1 and 2. We will define four measures of performance: total costs, expected shortage per cycle, fill-rate, and the Kullback–Leibler divergence. Finally, we present a multi-criteria decision analysis method using TOPSIS that occupies the previously indicated performance measures as criteria to evaluate DPUT and LTD modeling alternatives.

### *Total cost of continuous review with the shortage inventory model*

By carrying out the iterations shown in a previous subsection, we calculated the decision variables that minimize the total cost of continuous review with shortage inventory model $TC = G(Q*, r*)$, applied to Eq. (4) for each model in Tables 1 and 2. An inventory model is more effective when it results in a lower annual cost.

### *Expected shortage per cycle*

We used a previously explained performance measurement to obtain the value of the expected amount of shortage per cycle $(S(r*))$ given in Eq. (5), for each of the tested models in Tables 1 and 2. An inventory model is more effective when it results in a lower expected shortage per cycle.

### *Fill-rate*

(*Sobel, 2004*) defined the fill-rate of a supply system as: "the fraction of demand that is met from on-hand inventory, understanding that the satisfaction of the demand is restricted to the amount purchased and available". We calculated the Fill-rate for each statistical distribution shown in Tables 1 and 2 using Eq. (8):

$$Fill-rate = \frac{\int_o^{Q*} yf(y)\mathrm{d}y}{\int_0^{my} yf(y)\mathrm{d}y}, \tag{8}$$

where $f(y)$ are the PDFs that are shown in Table 1, and $my$ is the maximal DPUT for each of the evaluated distributions. An inventory model is more effective when its Fill-rate indicator value is closer to 1.

### *Kullback–Leibler divergence*

To determine the quality of the proposed PDF and LTD approximation in Table 2, we used the Kullback–Leibler divergence (*Cobb, 2004*):

$$Kullback-Leibler = \int_{-\infty}^{\infty} \log\left(\frac{f(x)}{\widetilde{f}(x)}\right) f(x)\mathrm{d}x, \tag{9}$$

where $f(\cdot)$ is the unknown true PDF and $\widetilde{f}(\cdot)$ its approximation. To calculate the Kullback–Leibler divergence show in (9), we used the kernel estimate to establish the true PDF (*Langseth et al., 2014*). Using Eq. (1), we computed the sequence $\{s_1, \ldots, s_n\}$ of $n$ LTD realizations (or data). According to the data, we defined the kernel estimate of the unknown

PDF of LTD $f_S(\cdot)$ using:

$$\widehat{f_S}(s) = \frac{1}{nh} \sum_{i=1}^{n} K\left(\frac{s_i - s}{h}\right), \ s \geq 0, \tag{10}$$

where $K(\cdot)$ is a kernel function satisfying $\int_0^\infty K(x)ds = 1$, $h$ a smoothing parameter and $s$ the point at which the PDF is estimated. Once Eq. (10) has been calculated, the Kullback–Leibler divergence for each LTD distribution established in Table 2 with support $[a, b]$ can be expressed as:

$$\text{Kullback– Leibler}' = \int_a^b \log\left(\frac{\widehat{f_S}(s)}{\widetilde{f_S}(s)}\right)\widehat{f_S}(s)ds, \tag{11}$$

where $\widehat{f_S}(s)$ is the kernel estimate and $\widetilde{f}(s)$ are the approximations proposed in Table 2. We selected one of the approximations with a smaller Kullback–Leibler value than the one calculated using Eq. (11).

### TOPSIS

This multi-criteria decision analysis analyzes the geometric distances between a chosen solution, the ideal solution, and the least suitable solution (*Yoon & Hwang, 1995*; *Aye, Gupta & Wanke, 2018*; *De Andrade, Antunes & Wanke, 2020*).

## NUMERICAL EXPERIMENTS AND ILLUSTRATIVE EXAMPLE

We used Algorithm 1 to generate intermittent DPUT and intermittent LTD. We performed 5000 repetitions or scenarios of DPUT samples using a ZITNO statistical distribution length $n = 30$, with the parameters $\mu_{NO} = 12$ and $\sigma_{NO} = 1$, a random proportion of zeros in the sample $(\nu)$, and an uniform generation in the interval [0,1]. This framework is applicable to each of the 5000 scenarios of the Montecarlo (MC) study, where we assumed an expected LTD value = 2 periods. For each scenario, we generated 200 DPUT data "jitter" simulated from the transition matrix of the Markov chain. Therefore, each LTD scenario had a length of 100 periods. For each scenario, we also generated uniform order costs (OC $\sim U[17; 140]$), holding costs (HC $\sim U[0; 0.68]$) and shortage costs (SC $\sim U[5; 50]$). These coefficients were chosen based on see Table 2 and Appendix C in (*Wanke, 2014*).

To model the intermittent DPUT we used the statistical distributions in Table 1, and to model LTD we used the statistical distributions shown in Table 2. Next, we estimated the parameters of normal DPUT and LTD probability distribution using the `fitDist` command of the `gamlss` package, and the `ets` and `crost` commands of the `tsintermittent` for Simple exponential smoothing and Croston's method in R software. We occupied both of these parameters as the OC, HC and SC coefficients to obtain $Q*, r*$, and $T_c$ values in Eq. (4) for each scenario. Interested readers can consult the codes attached to this work under the names "ZITNO.R" and "TOPSISSimulation.R".

### Evaluating inventory model performance measures using TOPSIS

In this subsection, we used the TOPSIS method to evaluate the performance of a continuous review inventory models with shortage. The DPUT/LTD modeling pairs were Simple

exponential smoothing / Simple exponential smoothing sum, Croston's /Croston's sum and ZITNO/ZITNOsum.

Figure 1 compare boxplots (A) to (O) of performance measures of TC, S(r) and Kullback–Leibler divergence, segmented in data groups formed according to the level of intermittency of the DPUT ($0 < v < 0.2$ , $0.2 < v < 0.4$, $0.4 < v < 0.6$, $0.6 < v < 0.8$ and $0.8 < v < 1$ ), between the proposal statistical distributions for results of 5000 scenarios of indicated inventory model. In this figure we have labeled Simple Exponential Smoothing as SES. Note that for all level of intermittency of the DPUT, except to $0 < v < 0.2$ the indicated performance measures of inventory model are lower using the ZITNO statistical distribution. We confirm this statement using the respective Kruskall-Wallis tests (results not shown). In the case of the fill-rate comparison there were no differences, and its median was always 1 in all the cases of intermittent levels of the DPUT and for all the statistical distributions studied (results not shown).

Table 3 shows the ranked % of TOPSIS order for each probability distribution model, segmented by intermittency level of the DPUT. Note that in all cases, the statistical distribution ZITNO get better performance in the inventory model of continuous review with shortage, occupying a multi-criteria evaluation of decisions using TOPSIS.

## Effect of variability and intermittent DPUT with ZITNO statistical distribution on total costs and *Q* and *r* decisions

This subsection discusses more in-depth how the parameter of variability ($\sigma_{NO}$) and the proportion of zeros ($v$) used to define the ZITNO statistical distribution affect the total costs and $Q$ and $r$ decisions. With this objective, we repeated the simulation scheme proposed in 'Numerical Experiments and Illustrative Example', but considered a more significant DPUT variability, setting the parameter at $\sigma_{NO} = 3$, and maintaining the parameter at $\mu_{NO} = 12$.

Figure 2 shows scatterplots between the proportion of zeros in the DPUT ($v$, labeled in x-axe as propDPUT) and TC, and $Q$ and $r$ decisions, in scenarios where $\sigma_{NO} = 1$ and $\sigma_{NO} = 3$.

We explored these relationships using classical multiple linear regression analysis, where to avoid the problem of multicollinearity we used standardized values of the independent variables $v$ and $\sqrt{v}$ (*Aiken, West & Reno, 1991*). Table 4 shows only the relationship between $r \sim v + \sqrt{v}$ ($\sigma_{NO} = 1$). In this model the adjusted R-squared is 0.90. Note that the regressor of $v$ is negative and significant, while the regressor of $\sqrt{v}$ is positive and significant. These relationships can be explained by the expression of $r = \mu_K \mu_{ZITNO} + z\sqrt{\sigma^2_{S_{ZITNOsum}}} = \mu_K \mu_{ZITNO} + z\sqrt{\sigma^2_{ZITNO}\mu_K} = \mu_K \mu_{ZITNO} + z\sqrt{\frac{1}{T+m}\sum_{t=T+1}^{T+m}(y_t - \mu_{ZITNO})^2 \mu_K}$, where $z$ is a security quantile of a standard normal distribution, and $\mu_{ZITNO} = (1-v)\int_{R_{y>0}} y\phi\left(\frac{y-\mu_{NO}}{\sigma_{NO}}\right)\frac{1}{\sigma_{NO}}dy$; with $y > 0$. Then, when $v$ decreases (there are more non-zero demands), $r$ increases to have enough stock to deal with this situation, while when $\sqrt{v}$ increases, the variance of the DPUT also increases, therefore it requires a larger safety stock and $r$ to have enough stock for this case.

0.8<v<1

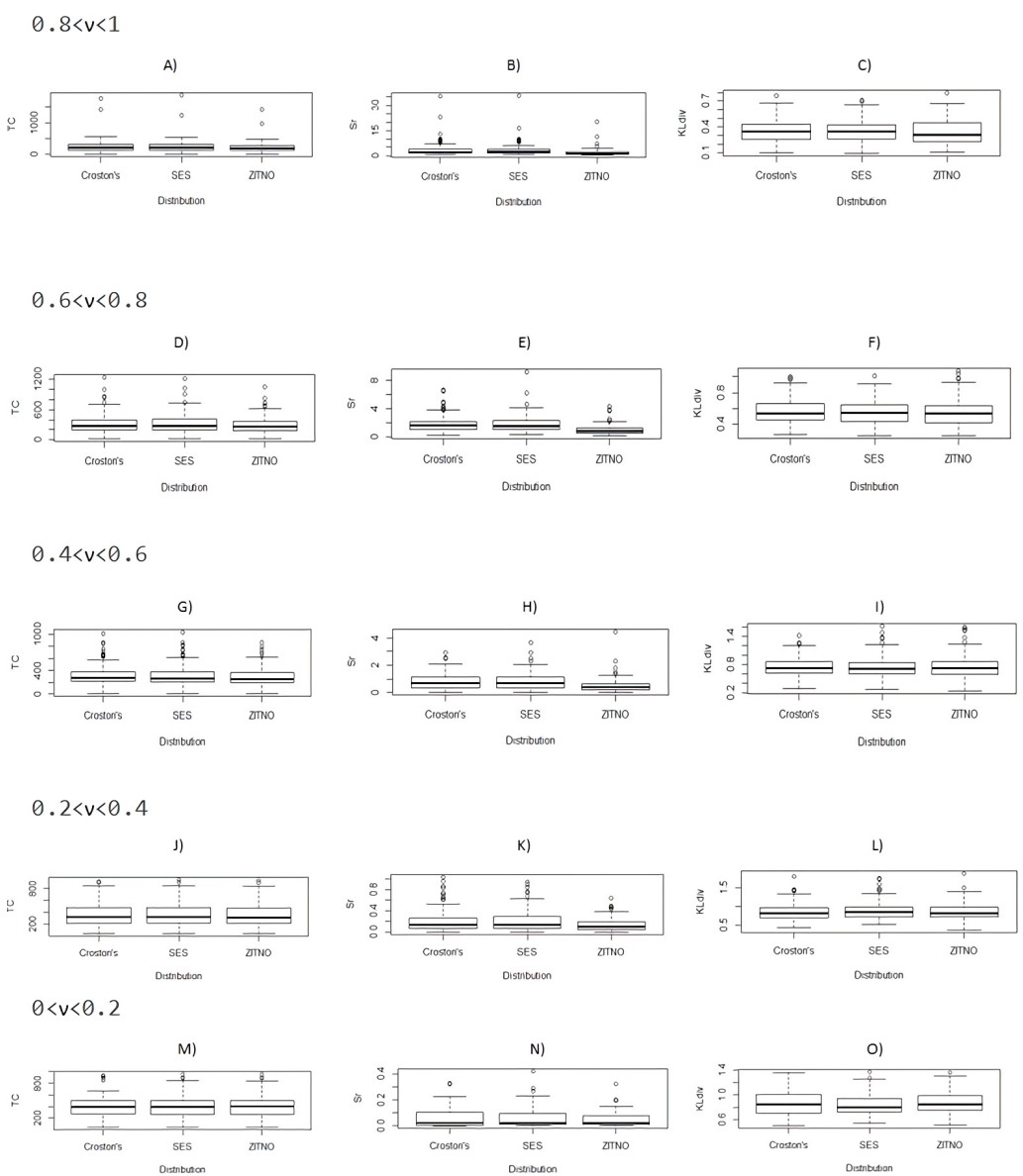

**Figure 1   Comparative boxplots (A) to (O) between proposal statistical distributions of TC, S(r) and Kullback–Leibler divergence segmented by v.**

Finally, we compared the differences between the optimal TCs, and the $Q$ and $r$ decisions where $\sigma_{NO} = 1$ and $\sigma_{NO} = 3$. Table 5 depicts descriptive measures such as mean, sd , interquartile range, and 0, 25, 50, 75 and 100-th quantiles of $r$ under both scenarios. We found significant differences for the Wilcoxon signed-rank test with continuity correction regarding $r$ decisions.

## Analyzing real data using an illustrative example

In order to show proposed method using real data, we selected a product experiencing intermittent demand from the inventory of a Chilean public pharmacy. First, we carried

Table 3 **Rank TOPSIS order (%) for statistical distributions compared.**

| Distributions | $v$ | Rank | TOPSIS | order (%) |
|---|---|---|---|---|
| | | 1 | 2 | 3 |
| ZITNO/ | $0.8 < v < 1$ | 97 | 2,74 | 0,26 |
| ZITNOSUM | $0.6 < v < 0.8$ | 99 | 0,85 | 0,15 |
| | $0.4 < v < 0.6$ | 94 | 5,03 | 0,97 |
| | $0.2 < v < 0.4$ | 98 | 1,96 | 0,04 |
| | $0 < v < 0.2$ | 98 | 1,9 | 0,1 |
| Simple exponential smoothing / | $0.8 < v < 1$ | 0,06 | 93,37 | 6,57 |
| Simple exponential smoothing sum | $0.6 < v < 0.8$ | 0,07 | 90,19 | 9,74 |
| | $0.4 < v < 0.6$ | 0,02 | 91,56 | 8,42 |
| | $0.2 < v < 0.4$ | 0,04 | 94,62 | 5,34 |
| | $0 < v < 0.2$ | 0,07 | 90,06 | 9,87 |
| Croston's / Croston's sum | $0.8 < v < 1$ | 2,94 | 3,89 | 93,17 |
| Croston's sum | $0.6 < v < 0.8$ | 0,93 | 8,96 | 90,11 |
| | $0.4 < v < 0.6$ | 5,98 | 3,41 | 90,61 |
| | $0.2 < v < 0.4$ | 1,96 | 3,42 | 94,62 |
| | $0 < v < 0.2$ | 1,93 | 8,04 | 90,03 |

out a statistical DPUT study using an adapted ZITNO statistical distribution. Second, we evaluated the performance of a $Q, r$ supply model with a DPUT shortage and ZITNO statistical distribution.

**Case presentation**. Public pharmacies in Chile do not base their supply practices on drug availability (*Rojas et al., 2019*). Instead, they manage and maintain a mix of inventory. As shown in (*Rojas et al., 2019*), models that factor in uncertain demands and shortages are useful for pharmacies. Currently, Chilean pharmacies manage their orders based on an annual needs planning with divided monthly orders and can be corrected up or down 20%, depending on the amount of inventory on hand. This method tries to comply with pharmaceutical safety recommendations, but suffers from supply decisions based on scientific criteria. This consequently increases total costs related to inventory management. Therefore, it is necessary to design a supply policy that minimizes involved costs, considers drug demands, and ensures that patients receive their treatments on time.

In this illustration, we carried out a statistical study of the DPUT for one product used in asthma treatments (called salbutamol) in an anonymous Chilean public hospital pharmacy. We proposed an optimized inventory system with reduced costs.

**Statistical study of the data set**. We studied a data set of the daily demand of salbutamol inhalators. The data set spanned 180 days. In order to study the temporal dependence of this data set, we looked at its autocorrelation function (ACF) and partial ACF (PACF) of DPUT, considering and not considering the null values (zeros). Figure 3 shows plots of the ACFs and PACFs. We detected a small partial autocorrelation when the null data (zeros) were included, which may be due to the fact that the article obeys a medical prescription

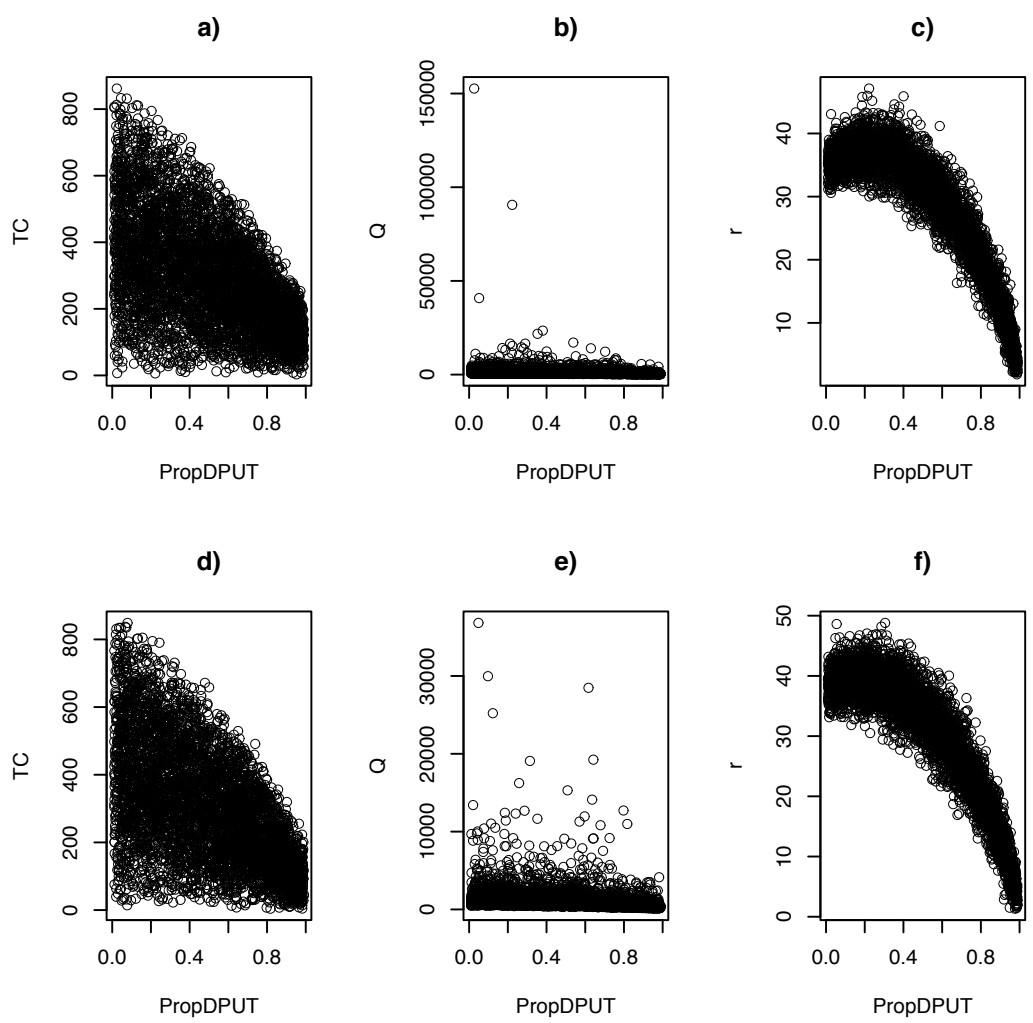

**Figure 2** Scatterplots: (A) TC $\sim v(\sigma_{NO} = 1)$, (B) Q $\sim v(\sigma_{NO} = 1)$, (C) $r \sim v(\sigma_{NO} = 1)$, (D) TC $\sim v(\sigma_{NO} = 3)$, (E) Q $\sim v(\sigma_{NO} = 3)$, and (F) $r \sim v(\sigma_{NO} = 3)$.

**Table 4** Relationship between $r \sim v + v \ (\sigma_{NO} = 1)$.

|  | Estimate | Std. Error | $t$ value | Pr(>|t|) |
|---|---|---|---|---|
| (Intercept) | 21.9553 | 0.2838 | 77.36 | 0.0000 |
| $v$ | −93.7066 | 0.8109 | −115.56 | 0.0000 |
| $\sqrt{v}$ | 80.2420 | 1.0007 | 80.19 | 0.0000 |

**Table 5** Descriptive measures of $r$ decisions (scenarios where $\sigma_{NO} = 1$ and $\sigma_{NO} = 3$).

| $r$ decisions | mean | sd | IQR | 0% | 25% | 50% | 75% | 100% |
|---|---|---|---|---|---|---|---|---|
| $r(\sigma_{NO} = 1)$ | 28.67 | 9.71 | 13.36 | 1.86 | 22.90 | 32.04 | 36.26 | 47.13 |
| $r(\sigma_{NO} = 3)$ | 30.28 | 10.28 | 14.88 | 1.34 | 23.50 | 33.66 | 38.38 | 48.81 |

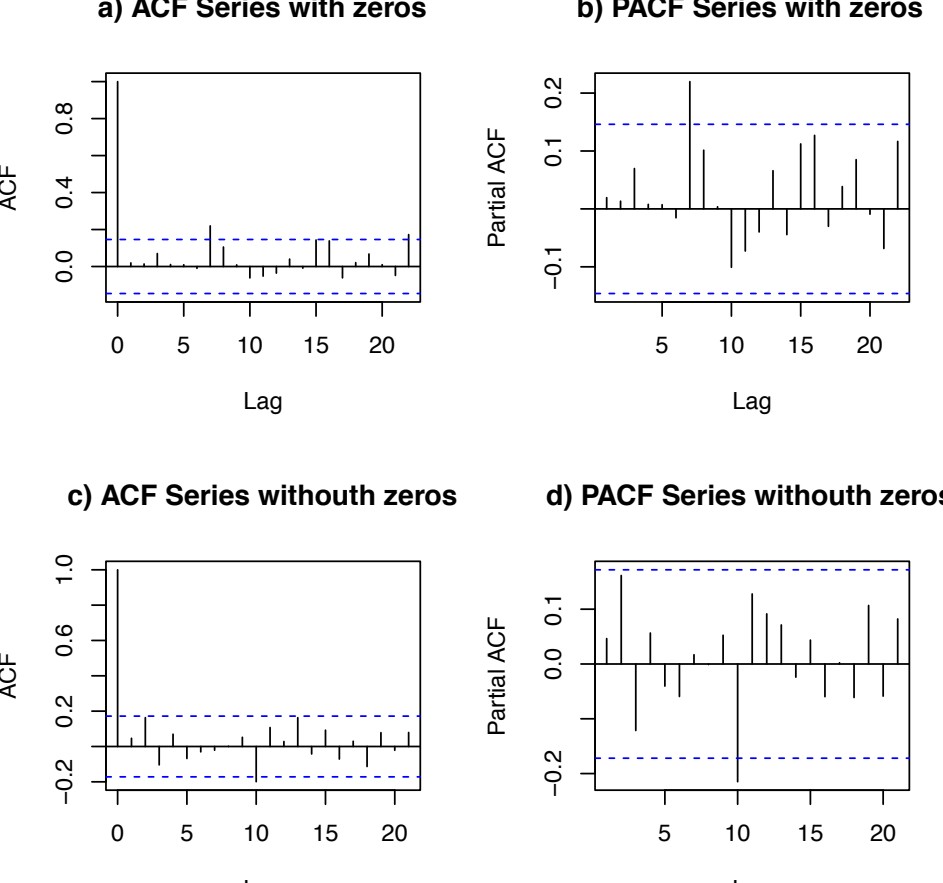

**Figure 3** ACF and PACF of the data set: with zeros (A and B) and without zeros (C and D).

**Table 6** Descriptive statistical indicators for the daily demand data set.

| Dataset | min | max | mean | median | sd | CV | CS | CK | n |
|---|---|---|---|---|---|---|---|---|---|
| With zeros | 0 | 31 | 14.6 | 18 | 9.78 | 0.67 | −0.54 | −1.18 | 180 |
| Without zeros | 9 | 31 | 20.21 | 20 | 4.18 | 0.21 | 0.058 | −0.05 | 130 |

every certain number of periods (10-day lag). In any case, the autocorrelation and partial autocorrelation was negligible.

Table 6 shows the size, minimum and maximum values, mean, median, sd, coefficient of variation (CV), coefficient of skewness (CS), and coefficient of kurtosis (CK) of the daily demand data set. The raw data of 'Analyzing real data using an illustrative example' is available to readers in supplemental files.

Figure 4 shows an empirical quantile–quantile plot to confirm the good standing of our proposed statistical distribution ZITNO DPUT model. Quantile–quantile plot is a graphical method for comparing two probability distributions by plotting their quantiles against each other. In this case, the proposed theoretical distribution (ZITNO) is compared

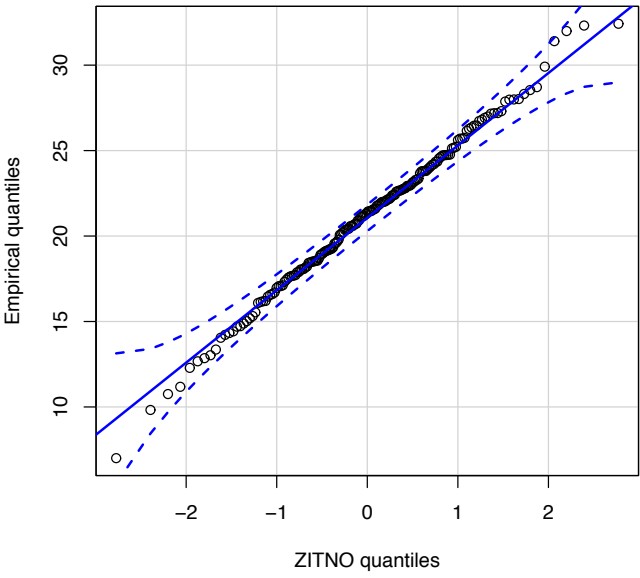

**Figure 4** **Quantile–quantile plot for statistical distribution proposal.**

**Table 7** **Parameters of the proposed statistical distribution to model DPUT and LTD and its competitors.**

| Statistical Distribution | Modeling Random variable | Parameters |
|---|---|---|
| ZITNO | DPUT | $\mu_{NO} = 14.6$ unit/day, $\sigma_{NO} = 9.78$ unit/day, $\nu = 0.3$ |
| Simple exponential smoothing | DPUT | $\mu_{SES} = 14.61$ unit/day, $\sigma_{SES} = 9.75$ unit/day |
| Croston's | DPUT | $\mu_{CROST} = 15.35$ unit/day, $\sigma_{CROST} = 10.12$ unit/day |
| ZITNOsum | LTD | $\mu_{NO}\mu_K = 29.2$ unit/day, $\sigma_{ZITNO}\sqrt{\mu_K} = 18.83$ unit/day, $\nu_s = 0.1$ |
| Simple exponential smoothing sum | LTD | $\mu_{SES}\mu_K = 29.22$ unit/day, $\sigma_{SES}\sqrt{\mu_K} = 13.78$ unit/day |
| Croston's sum | LTD | $\mu_{CROST}\mu_K = 30.7$ unit/day, $\sigma_{CROST}\sqrt{\mu_K} = 14.31$ unit/day |

with respect to the empirical distribution, where the points should ideally approach a diagonal line. Since all values were within confidence bands, we propose that the statistical distribution correctly fits the data.

Table 7 shows the parameters calculated using our proposed statistical distribution and its competitors.

**Proposed statistical distribution inventory model performance.** We considered the following costs involved in the application of a $Q, r$ inventory model with shortage: HC= 0,042 USD\$/(unit\*year), OC= 0,86 USD\$/order, SC= 0,33 USD\$/cycle, and constant LTD = 2 days. Table 8 shows the performance measures relative to Q*, r, TC, S (r), Fill-rate, and Kullback–Leibler divergence when applying an $Q, r$ inventory model with shortage and the

**Table 8  $Q, r$ model with shortage performance measures using the proposed statistical distribution to model DPUT/LTD.**

| Modeling DPUT/LTD | Q* (unit) | r (unit) | TC (USD) | S(r) (quantity of shortage/ cycle) | Fill-rate | Kullback–Leibler divergence |
|---|---|---|---|---|---|---|
| ZITNO/ ZITNOsum | 1,147 | 32 | 393.82 | 0 | 1 | 0.78 |
| Simple exponential smoothing / Simple exponential smoothing sum | 1,372 | 35 | 468.34 | 0 | 1 | 0.82 |
| Croston's /Croston's sum | 1,406 | 36 | 480.05 | 0 | 1 | 0.80 |

proposed statistical distributions for the DPUT/LTD modeling. Note that all performance measures using ZITNO/ZITNOsum favored our intermittent demand model.

## DISCUSSION

We adapted our Algorithm 1 from (*Willemain, Smart & Schwarz, 2004*), for the use of ZITNO and ZITNOsum distributions to model DPUT and LTD, respectively.

Our model optimizing of the annual total cost of expected inventory with shortage results in lower total costs and smaller shortages per cycle in almost all cases compared to traditional methods used to model intermittent demand such as simple exponential smoothing and its variant, Croston's method. Table 3 shows that the ZITNO/ZITNOsum statistical distribution method performs better than the traditional slow-moving inventory models when modeling intermittent DPUT and LTD. Here, we must acknowledge that the standard methods also achieved good Fill-rate with a satisfaction of the DPUT. However, in most cases, our proposal achieved lower total costs and smaller non-supplied quantities than traditional methods. Our proposed method was effective regardless of the number of zeros contained in the DPUT data samples. The simple exponential smoothing and Croston's method approaches have been extensively employed in intermittent demand forecasting (*Balugani et al., 2019*). However, they lack the properties of a statistical distribution, so they generally show low performance measures when used in stochastic inventory models, such as the one studied in this work.

Once we confirmed that the ZITNO statistical distribution achieved good yields in the considered inventory model, we studied how the parameter of variability and the proportion of zeros that defined this statistical distribution affected total costs, $Q$ and $r$ decisions, and possible connections. The most important relationship found was between the proportion of zeros in DPUT, which shows the degree of intermittency of this variable, the square root of this same parameter and the reorder point decisions. This relationships were explained by a multiple linear regression model. At first glance, the low intermittency of DPUT has a positive proportionality related with the square root of the parameter of proportion of zeros in demand ($\sqrt{v}$), and later it suffers a decay by increasing the intermittency of the DPUT ($v$). This behavior is important to decision-makers that need to consider the degree of DPUT intermittency for their reorder point decisions. We confirmed that the parameter of variability of the ZITNO statistical distribution positively correlated with reorder point decisions. That is, as the variability of the non-zero DPUT increases, the

reorder points must also increase. The previous conclusion was logical since this indicator can protect shortages against scenarios of greater variability. However, it does not alter the ordered quantities or the total costs of the inventory policy.

In our study, we looked at actual data from a real case study to corroborate our method's performance.

We think that the use of ZITNO statistical distribution is especially suitable for intermittent demand due to its capability for modeling non-zero and zero demand cases. We tested our method by calculating indicated statistical distribution, and achieved good LTD distribution adjustment results. We also obtained good results when the non-zero data were slightly asymmetrical and when the zero values of the DPUT showed a high proportion. Our main objective was to create models as close to reality as possible, but we acknowledge that this topic of study is an area of ongoing research that need more empirical results in future research. In this context, it is necessary to study the adaptation to skewness and kurtosis of the non-zero data of an intermittent demand in diverse stochastic inventory models, for this and other mixture statistical distributions, because these characteristics of the probability distributions could directly affect the results obtained in $r$ and $S(r)$.

Another important limitation to point out is that the busy optimization method is for each item or product in a particular way. For this reason, it would be important to consider multi-product stochastic programming in future research considering our proposed ZITNO and ZITNOsum statistical distributions. In the future, we plan to address some limitations shown in this study, such as the assumption of constant LT, which we used to model the LTD as a sum of DPUT.

## CONCLUSION

In this paper, we developed a new methodological framework for intermittent demand modeling.

We were able to generate an LTD jitter in the case of an intermittent DPUT. We used numerical experiments to show that our proposed statistical distributions ZITNO and ZITNOsum leads to better results in a continuous revision inventory model with shortage. In particular, we used the multi-criteria TOPSIS method across multiple scenarios with different proportions of zeros in the DPUT and cost of ordering, storing, and shortage parameters.

In slow-moving items modeled by our proposal of statistical distribution, decisions $Q$ and $r$ are affected by the level of intermittent demand. Both decrease but not proportionally in the case of the decision of $r$, because the proportion of zeros in the DPUT is a parameter that affects the variability of the LTD.

### Funding

This work was supported by Fondecyt initiation project code: 11190004, Chile. The funders had no role in study design, data collection and analysis, decision to publish, or preparation of the manuscript.

### Grant Disclosures

The following grant information was disclosed by the authors:
Fondecyt initiation project code: 11190004, Chile.

### Competing Interests

The authors declare there are no competing interests.

### Author Contributions

- Fernando Rojas conceived and designed the experiments, performed the experiments, analyzed the data, performed the computation work, prepared figures and/or tables, authored or reviewed drafts of the paper, and approved the final draft.
- Peter Wanke conceived and designed the experiments, authored or reviewed drafts of the paper, and approved the final draft.
- Giuliani Coluccio and Juan Vega-Vargas performed the experiments, analyzed the data, authored or reviewed drafts of the paper, and approved the final draft.
- Gonzalo F. Huerta-Canepa performed the experiments, analyzed the data, performed the computation work, authored or reviewed drafts of the paper, and approved the final draft.

### Data Availability

A total of 5000 scenarios simulated of DPUT with diverse level of intermittency and actual data of an illustrative example are available in the Supplemental Files.

### Supplemental Information

Supplemental information for this article can be found online at http://dx.doi.org/10.7717/peerj-cs.298#supplemental-information.

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
