# Peer review of "Managing slow-moving item: a zero-inflated truncated normal approach for modeling demand"

_PeerJ Computer Science, doi:10.7717/peerj-cs.298_

## Round 0.1 · original submission · Major Revisions

All reviewers raised several suggestions which the authors should address.

·

Basic reporting

(a) Some examples where the language could be improved include lines 95,99,281 and so on.
(b) Almost all pictures are blurry, and some of them even affect the judgment on the experimental results.
(c) The layout of the table is not good and worthy of improvement.
(d) It's better to add figures to illustrate the execution of the proposed method.

Experimental design

(a) The comparison experiment on simulation data is clear, however, there is no compare on real data in section 3.3. The simulated data conforms to the article's data distribution assumption, and the results are certainly not bad. It is persuasive to beat other methods on real data sets.
(b) In practical applications, as merchants will not only sell one product, the inventory management of multiple products instead of single product should be analyzed.

Validity of the findings

(a) Please correct the mistakes in Table 3 RankTOPSISorder(%) .
(b) The raw data in section 3.3 is needed.

Additional comments

This paper adapted existing work for slow-moving management, using ZINO and ZINOsum distributions to model DPUT and LTD. The authors did a lot of work, but there are still some shortcomings. The aspects that need improvement are listed above.

Reviewer 2 ·

Basic reporting

- It is important to clarify that the mixture distribution defined here is a zero-inflated Truncated Normal - the Normal distribution component's domain is defined in positive Real only (at least from the code).
- The extensive use of acronyms is a bit distracting: I recommand keeping only DPUT, LTD, TOPSIS and ZINO, and spell out random variable (RV), exponential smoothing (SES), and Croston's (CROST) / Croston's sum (CROSTsum).
- Line 90-91 since K is a fixed value, there is no need to describe the mean and variance.

Experimental design

In general, the author should improve the supplementary code:
- in ZINO.R it is better to use the standard Normal density function dnorm in dZINO instead of your own implementation.
- A bit of formating would help the readability of the R codes (using tools like https://styler.r-lib.org/index.html).
- If I understand correctly, Algorithm 1 is implemented in jitter.R - It would be great that this is made more clear either in the name of the file or the comment in the code.
- In the TOPSIS simulation code, the optimization with the iterative process basically copies and paste of the same lines multiple times - these should be rewritten into a for loop or a while loop

Validity of the findings

- Table 3 is badly formatted and quite challenging to read. It would be ideal if it could be presented as a figure.
- Figure 1 and 2 needed to be better formatted so every subplot are consistent in size, also you should increase the resolution and font size.
- I am not sure I understand the B-spline analysis, and the phrasing that nonlinear correlation between the r and v could be "explained" by B-spline is problematic and not theoretical motivated. To me it is more an exploratory analysis on the underlying nonlinear correlation.

Additional comments

The current MS introduced a slow-moving item management model (intermittent DPUT) that takes advantage of the flexibility of mixture distribution (i.e., zero-inflated normal) to better distinguish between non-zero and zero DPUTs. The author shows the advantage of the proposed method compare to models with other distribution for DPUT and LTD using TOPSIS.

The MS is well written, and I only have some minor comments.

Reviewer 3 ·

Basic reporting

This paper presents Managing slow-moving item using of intermittent demand per unit time and lead time demand.

I would suggest to accept this submission after getting the following major modifications incorporated:
- it is suggested to get the draft proof-read by a language expert to improve the writing of the draft
- the reference list should be updated with more recent references from strong publication venues
- in section 2, the subsections should be further divided into the following subsections: 2.1, 2.2, 2.3, and 2.4
- in subsection 3.2 and 3.3, Figure 1, 2, and 3 should be redrawn with better visualization
- in subsection 3.1 and 3.3, Table 3 and 7 are mixed up, it should be redrawn

Experimental design

- in subsection 3.3, Figure 3 shows a very straight data line through the plot, experiments should be done with more data to show
- subsections 3.1 and 3.2 should be discussed in details with more experiment setting
- in subsection 3.4, more experimental results should be added to the proposed statistical distribution inventory model

Validity of the findings

Novelty, clarity of expression, and background and related work are good but it lacks the experimental results.

---

## Round 0.2 · Minor Revisions

Please address the remaining comments.

·

Basic reporting

A. All pictures are still blurry. Even the original eps version is not clear enough to view. Please enhance the resolution to prove originality.
B. The same items in Table 3 can be merged.

Experimental design

no comment

Validity of the findings

no comment

Additional comments

If the resolution of figures was improved, the paper can be accepted.

Reviewer 2 ·

Basic reporting

no comment

Experimental design

The author should went through the code again to make sure naming is consistent since ZINO is now renamed to ZITNO.

Validity of the findings

no comment

Additional comments

The author have addressed all my comments.

Reviewer 3 ·

Basic reporting

The revised version is in improved form but I would suggest the following modification:
- update the literature section from the past 3 years papers
- Figures 1 and 2 in sections 3.1 and 3.2, respectively, should have clear labels

Experimental design

In the revised manuscript, the response from the authors is satisfactory.

Validity of the findings

Section 4 should address possible future research problem domains.

Additional comments

Overall, the authors have responded satisfactorily and the manuscript is now in an improved form. I would suggest enriching section 4 by adding some comments on low performing traditional methods as well. Moreover, in future work, some concrete directions should be added to help the reader what kind of research problems in this domain are still unaddressed.

---

## Round 0.3 · accepted · Accept

The authors have addressed the remaining comments.